# TIME TRAVEL IN LLMS: TRACING DATA CONTAMINATION IN LARGE LANGUAGE MODELS

**Shahriar Golchin**[*]**, Mihai Surdeanu**
Department of Computer Science, University of Arizona
`{golchin,msurdeanu}@arizona.edu`

## ABSTRACT

Data contamination, i.e., the presence of test data from downstream tasks in the training data of large language models (LLMs), is a potential major issue in measuring LLMs' real effectiveness on other tasks. We propose a straightforward yet effective method for identifying data contamination within LLMs. At its core, our approach starts by identifying potential contamination at the *instance level*; using this information, our approach then assesses wider contamination at the *partition level*. To estimate contamination of individual instances, we employ "guided instruction:" a prompt consisting of the dataset name, partition type, and the random-length initial segment of a reference instance, asking the LLM to complete it. An instance is flagged as contaminated if the LLM's output either exactly or nearly matches the latter segment of the reference. To understand if an entire partition is contaminated, we propose two ideas. The first idea marks a dataset partition as contaminated if the average overlap score with the reference instances (as measured by ROUGE-L or BLEURT) is statistically significantly better with the completions from guided instruction compared to a "general instruction" that does not include the dataset and partition name. The second idea marks a dataset partition as contaminated if a classifier based on GPT-4 with few-shot in-context learning prompt marks multiple generated completions as exact/near-exact matches of the corresponding reference instances. Our best method achieves an accuracy between 92% and 100% in detecting if an LLM is contaminated with seven datasets, containing train and test/validation partitions, when contrasted with manual evaluation by human experts. Further, our findings indicate that GPT-4 is contaminated with AG News, WNLI, and XSum datasets.[1]

## 1 INTRODUCTION

The rise of Transformer networks (Vaswani et al. 2017) has spurred the development of large language models (LLMs), marking a new epoch in Natural Language Processing (NLP). This shift has led to an extensive range of LLMs (Touvron et al. 2023a;b; Biderman et al. 2023; Köpf et al. 2023; Chung et al. 2022; Penedo et al. 2023, inter-alia) which excel in various professional and academic benchmarks (Bang et al. 2023; Bubeck et al. 2023). Their superior performance is primarily attributed to the massive web data consumed by these billion/trillion-parameter LLMs during training. However, the impressive LLM performance observed on many downstream tasks (e.g., summarization, natural language inference, text classification) may be inflated due to *data contamination*, i.e., the presence of test data from these downstream tasks in the pre-training data of LLMs. Guaranteeing lack of contamination is not trivial due to two potential sources of contamination: directly from ingesting the official version of a dataset (easier to control), and indirectly through duplicated data found somewhere on the web (nearly impossible to control).[2] The potential of data contamination is especially relevant for closed models such as the GPT-3/3.5 family (Brown et al.

---

[*]Corresponding author.

[1]See the paper's repo at `https://github.com/shahriargolchin/time-travel-in-llms`.

[2]While dataset licensing reduces indirect contamination to a certain extent, it does not eliminate it. For example, websites such as the Hugging Face page for datasets (Wolf et al. 2020) currently host copies of the OntoNotes (Weischedel et al. 2013) and CoNLL-2003 (Tjong Kim Sang & De Meulder 2003) datasets, despite the fact that their respective licenses prohibit it.

2020) and GPT-4 (OpenAI 2023; Bubeck et al. 2023), and, needless to say, raises questions on the validity of evaluations and benchmarks conducted so far (Chang et al. 2023; Zhu et al. 2023; Bordt & von Luxburg 2023; Ray 2023; Penedo et al. 2023).

To address this issue, we propose an inexpensive and robust approach to detect data contamination for a given dataset partition automatically. Importantly, our approach functions under two realistic assumptions: **(a)** we lack direct access to the pre-training data of the LLMs, and **(b)** we have limited computational resources. Intuitively, our method starts by identifying potential contamination in *individual instances* that are drawn from a small random sample of the corresponding dataset partition (we use samples of 10 instances in this work). Using the information obtained from individual instances, our approach then assesses if an *entire dataset partition* is contaminated.

More formally, to identify contamination of individual instances, we employ a "guided instruction:" a prompt that integrates distinct identifiers from the source dataset from which the reference instance originates. Such information includes the dataset name, its partition (e.g., train, test, or validation), and a randomly selected initial portion of the reference instance, complemented by its label when relevant. With these signals in the prompt, we instruct the LLM to finish the given partial instance. Using these generated *individual* completions, we propose two heuristics to estimate if an *entire* dataset partition is contaminated. The first heuristic states that a partition is likely to be contaminated if the average overlap score between generated completions and reference instances (as measured by ROUGE-L (Lin 2004) and BLEURT (Sellam et al. 2020)) observed with the guided instruction is statistically significantly larger than the one measured with a "general instruction," which does not include the dataset and partition name. The second heuristic labels a partition as contaminated if a classifier based on GPT-4 with few-shot in-context learning (ICL; Brown et al. (2020)) marks at least one generated completion as an exact match with the reference instance or at least two generated completions as near-exact matches, where near-exact match indicates a completion that exhibits considerable semantic and lexical alignment with the reference instance.

The primary contributions of this paper are as follows:

**(1)** We propose a novel data contamination detection method for LLMs that is inexpensive and robust. As indicated above, our method combines a "guided instruction" to complete partial instances randomly drawn from the investigated dataset partition and several heuristics to generalize from instance- to partition-level contamination decisions.

**(2)** We evaluate our proposed methods in 28 distinct scenarios. These scenarios are created by two state-of-the-art LLMs: GPT-3.5 and GPT-4, and span seven datasets for classification, summarization, and natural language inference (NLI) tasks. The rationale behind the 28 scenarios is that for each dataset, we separately explore potential data contamination in the train and test splits (or the validation set, in cases where the labeled test set is not publicly available). Our evaluation indicates that our best method is the one that uses guided instruction to complete partial instances, and the one that evaluates these completions by the GPT-4 few-shot ICL classifier, achieving 92%–100% accuracy compared to contamination labels assigned by human experts for dataset partitions.

**(3)** Our analysis indicates that GPT-4 showed evidence of contamination within the test partitions of AG News (Zhang et al. 2015), WNLI (Wang et al. 2018), and XSum (Narayan et al. 2018) datasets. These findings support the observation that data contamination is a serious issue that must be considered in downstream evaluations when using LLMs.

## 2 RELATED WORK

Despite its importance, the topic of data contamination is not as thoroughly examined as its closely related field, data memorization (Carlini et al. 2023; Kandpal et al. 2022; Carlini et al. 2021; Razeghi et al. 2022). Among the limited investigations focusing specifically on data contamination in LLMs, we find notable examples in Radford et al. (2019) and Brown et al. (2020) on GPT-2 and GPT-3, respectively. They used high-order $n$-grams (e.g., 13-gram) to detect overlapping content between the pre-training data and the evaluation dataset. Most research subsequent to Brown et al. (2020) adopted similar methods for detecting data contamination (Touvron et al. 2023b; Du et al. 2022; Chowdhery et al. 2022; Wei et al. 2021), and most recently, substring matching for GPT-4 (OpenAI 2023). However, the scope of existing research has been predominantly confined to model providers, and it encounters specific limitations, particularly when applied to closed-source LLMs. These

limitations primarily involve the need for access to pre-training data (Brown et al. 2020; Du et al. 2022; Wei et al. 2021), the requirement for substantial computational resources (Touvron et al. 2023b), or the need for extensive manual labor (Chowdhery et al. 2022). Our approach aims to overcome these hurdles, enabling the assessment of data contamination in scenarios where *the pre-training data is either not openly accessible* or *when significant computational hardware is not available* despite having access to the pre-training data.

Our paper is closest in spirit to the work of Sainz et al. (2023), who also detected contamination when access to the pre-training data is not available. This effort prompted ChatGPT, particularly when GPT-3.5 is its base model, to generate the first instances from different dataset partitions. The underlying assumption here is that if an LLM can reproduce dataset instances, it must have been trained using that particular split. However, our research shows that this method can be unreliable and subject to failure. Such failures can result either from the sparsity introduced by the request to reproduce the first instances of a dataset split or from the inability to bypass the safety filters set by the model provider when the model is asked to generate copyrighted content like dataset instances. Throughout this paper, we refer to this approach as "ChatGPT-Cheat?," taking inspiration from the title of the referenced blog post.

## 3   APPROACH

In our approach, we operate under two core assumptions: (1) lacking direct access to the pre-training data of the LLMs, and (2) having limited computational resources. Given these premises, our detection strategy for data contamination is anchored by two pivotal insights. First, we examine individual instances within a dataset partition to spot *contamination at the instance level*. Second, given that LLMs are pre-trained on large-scale data, detecting contaminated instances can act as a *signal of broader contamination*. As a result, the associated partition can be labeled as being leaked to the LLM's pre-training data.

To discern contamination at the instance level, we focus on replicating instances by the LLM. In this context, exact replicas of instances serve as red flags for contamination in the corresponding partition. Note that, due to the inherent probabilistic behavior of LLMs, achieving perfect replicas is not always possible even when contamination is certain. Nevertheless, instances that are closely replicated have a twofold function: while they can offer insightful indications of potential contamination, the fact that many datasets draw from web-based sources implies that partial replicas can also arise by happenstance. This overlap introduces uncertainty in drawing a definitive conclusion about the underlying partition. Thus, it is essential to check for consistent and significant signs of contamination within the partition.

In the following sections, we first elaborate on our method and the necessary components to compel LLM into reproducing dataset instances. We then delve into the procedure for evaluating the contamination status of existing LLMs for an entire partition based on these instances. Furthermore, leveraging the fine-tuning option offered by OpenAI for the GPT-3.5 base model, we undertake a study in which we intentionally contaminate the GPT-3.5 base model with partitions that our method detected as uncontaminated. Subsequently, we subject the contaminated GPT-3.5 to our technique, further showcasing our method's effectiveness in pinpointing data contamination within LLMs.

### 3.1   DETECTING INSTANCE-LEVEL CONTAMINATION

#### 3.1.1   COMPONENTS TO MEASURE INSTANCE-LEVEL CONTAMINATION

To gauge instance-level contamination, we utilize two distinct methods: the first leverages BLEURT and ROUGE-L scores, while the second draws on few-shot ICL prompting with GPT-4. Each method employs particular components; however, the first two—guided instruction and the next token prediction mechanism—are shared. The third component—general instruction—is exclusive to the first method. For both methods, we begin our process by steering the LLM towards the (potentially contaminated) dataset partition using guided instruction that integrates the dataset name, partition of interest, and the random-length initial segment of a randomly selected instance and its label if it is available. The LLM is then instructed to complete it. For the first method, we repeat this step using general instruction that omits the dataset and partition name. An example of a guided versus a general instruction is depicted in Figure 1. We detail all the required components below.

> **Instruction:** You are provided with Sentence 1 from the validation split of the WNLI dataset. Finish Sentence 2 as appeared in the dataset. Sentence 2 must exactly match the instance in the dataset.
>
> **Sentence 1:** The dog chased the cat, which ran up a tree. It waited at the top.
>
> **Label:** 1 (entailment)
>
> **Sentence 2:**
>
> `The cat waited at the top.`

> **Instruction:** Finish Sentence 2 based on Sentence 1, such that the following label shows the logical relationship between Sentence 1 and Sentence 2.
>
> **Sentence 1:** The dog chased the cat, which ran up a tree. It waited at the top.
>
> **Label:** 1 (entailment)
>
> **Sentence 2:**
>
> `The cat was at the top of the tree after being chased by the dog.`

Figure 1: An example of a guided (left) and general (right) instruction employed for a paired-instance dataset. In this example, using GPT-4, the guided instruction results in an exact match, whereas the general instruction does not.

**(1) Guided Instruction—A Means to Navigate the LLM's Output.** By employing instruction-tuning on top of causal language modeling (CLM; Vaswani et al. (2017); Radford et al. (2018)), LLMs can be guided by human directives (Wei et al. 2022; Sanh et al. 2022; Chung et al. 2022). This serves as a tool for controlling the LLM's output using natural language. Thus, we form guided instruction such that it *incorporates the dataset and split name in the input prompt*, thereby directing the LLM towards the underlying dataset split. A comprehensive list of all the instructions used in this study for different tasks/datasets can be found in Table 5 in Appendix A.

**(2) Next Token Prediction—A Means to Unravel Data History.** Primarily, data contamination occurs during the CLM pre-training phase since it constitutes the largest part of training in LLMs and utilizes web data. Without instruction tuning, an LLM only attempts to complete an input prompt based on data seen during the CLM pre-training phase (Ouyang et al. 2022). Notable models that exhibit this behavior include GPT-2 and GPT-3. We, therefore, employ the next token prediction mechanism to trace data history. In particular, we feed the model the variable-length initial segment of a dataset instance, chosen randomly from a particular split, prompting it to finish the partial instance. For labeled instances, we integrate the corresponding labels in the input prompt. This reflects that if an instance was ingested during the LLM's pre-training, its label was ingested too.[3]

For paired-instance datasets, we present the model with the initial sentence and its corresponding label. In the case of single-instance datasets, instances with multiple sentences are arbitrarily cut at the end of a complete sentence, whereas for instances containing a single (long) sentence, a random sentence fragment is eliminated. Finally, the LLM is tasked with finishing the provided initial part. Figure 1 shows this process for a paired-instance dataset.

Therefore, once a contaminated LLM is prompted with guided instruction, its output should mirror the subsequent segment of the reference instance under the guidance of the dataset and split name.

**(3) General Instruction—An Alternative Facet of Causal Language Modeling.** We formulate the general instruction to measure the impact of the guidance given in the guided instruction. This general instruction only requests the completion of the partial instance without specifying the dataset or its partition. As a result, when using this instruction, the generated sequence solely relies on the CLM pre-training phase, akin to autoregressive models without instruction tuning. This enables us to establish a baseline for generated random replicas and assess how much the guided instruction influences the LLM-generated part of the input partial instance. We assess this influence in terms of overlap, semantics, and structural similarity with the reference instance. This analysis is crucial as even when the output of LLM does not perfectly match the reference instance, it still enables us to detect potential signs of contamination.

### 3.1.2 MEASURING INSTANCE-LEVEL CONTAMINATION

We introduce two methods for measuring contamination at the instance level:

---

[3]Incorporating labels in the input prompt is essential to account for false positives when generating downstream completions. Illustrations of the impact of label integration on downstream completions are provided in Table 6 in Appendix B.

**BLEURT & ROUGE-L:** To quantify the overlap between the completions—produced under both guided and general instructions—and reference instances, we employ two metrics: ROUGE-L (Lin 2004) and BLEURT (Sellam et al. 2020). While ROUGE-L assesses lexical similarity, BLEURT gauges the semantic relevance and fluency of the resulting sequence with respect to the reference instance. Instance-level contamination is detected if the average overlap scores from either metric, when applied to completions from the guided instruction, exceed those from the general instruction.

**GPT-4 Evaluation:** While both BLEURT and ROUGE-L quantify the overlap between the generated and reference instances, they fall short of pinpointing near-exact matches. To bridge this gap, we adopt few-shot ICL prompting (Brown et al. 2020) to dictate the detection of exact/near-exact matches based on human judgments (see Section 4: Human Evaluation for our definition of a near-exact match). Specifically, this method includes a few representative examples of exact and near-exact matches—sourced from human evaluations—in the prompt, which are used to assess all other generated completions. We chose GPT-4 for this task as it requires no specialized prompting technique (Bubeck et al. 2023), enhancing the reliability of its results. A visual representation of the few-shot ICL prompt used in our study can be seen in Figure 3 in Appendix C. Also, detailed examples, including their ROUGE-L and BLEURT scores, as well as both human and GPT-4 few-shot ICL evaluations, are listed in Table 7 in Appendix D.

## 3.2 DETECTING PARTITION-LEVEL CONTAMINATION

To generalize from instance-level contamination to partition-level discrete decisions (i.e., the partition is/is not contaminated), we take advantage of two observations:

**Idea 1:** *A dataset is likely to be contaminated if the average overlap score with the reference instances (as measured by ROUGE-L and BLEURT) observed with completions from the guided instruction is significantly larger than the one measured with the completions from the general instruction.* The motivation behind this idea is that since the only difference between the two instructions is that the guided instruction contains the dataset and partition name as guidance, the improvement can only be explained by contamination.

**Idea 2:** *A dataset is likely to be contaminated if GPT-4 using few-shot ICL prompting detects at least one exact match or at least two near-exact matches.* The intuition behind this idea is that even a small contaminated part of the sample of instances is likely indicative of a larger dataset partition leak. While the presence of an exact match among replicas generated by LLM is a clear sign of contamination, the approach to handling exact or near-exact matches—and deciding the number of such matches that indicates broader contamination—can be tailored depending on specific research objectives. In this paper, we intuitively establish the above-mentioned criterion to extrapolate from the instance-level to the partition-level contamination. An empirical validation of our approach is also provided in Section 3.3.

We propose two algorithms, each implementing one of these ideas respectively.

**Algorithm 1:** A dataset partition is labeled as contaminated if the average overlap score (as provided by BLEURT and ROUGE-L) between the reference instances and generated texts with the guided instruction on a sample of ten instances is statistically significantly better than those produced by general instruction under a non-parametric bootstrap resampling test.[4]

The advantage of this algorithm is that it is non-parametric, i.e., we do not need to decide on an arbitrary threshold on the ROUGE-L or BLEURT scores to indicate contamination. However, its drawback is that even a significant increase in overlap may still come from generated instances that a human would not consider an exact or near-exact match. Algorithm 2 addresses this limitation.

**Algorithm 2:** A dataset partition is labeled as contaminated if GPT-4 with few-shot ICL prompting flags at least one generated completion as an exact match or a minimum of two completions as near-exact matches within a sample of ten instances. All completions in this setting are generated solely by guided instruction.

We evaluate both these algorithms in Section 5.

---

[4]Details of our bootstrap resampling method can be found in Appendix E.

---

**Instruction:** You are provided with Sentence 1 from the train split of the RTE dataset. Finish Sentence 2 as appeared in the dataset. Sentence 2 must exactly match the instance in the dataset.

**Sentence 1:** Twelve of Jupiter's moons are relatively small and seem to have been more likely captured than to have been formed in orbit around Jupiter.

**Label:** 0 (not entailment)

**Sentence 2:**

---

`GPT-3.5:` The formation of Jupiter's twelve relatively small moons is more likely due to their capture rather than being formed in orbit around Jupiter.

---

`Contaminated GPT-3.5:` Jupiter has twelve moons.

---

Figure 2: An example of an exact match generated by the GPT-3.5 contaminated with the train split of the RTE dataset versus an inexact match generated by the GPT-3.5 base model, both under the same guided instruction. This example is one of the training instances used during contamination.

Table 1: Results after introducing intentional contamination to the GPT-3.5 base model using guided instruction. A tick (✓) indicates the identification of at least one exact replica from the training instances used for contamination by our top-performing method (Alg. 2: GPT-4 ICL) and human evaluation.

| Method | AG News | RTE | XSum |
|---|---|---|---|
| Alg. 2: GPT-4 ICL | ✓ | ✓ | ✓ |
| Human Evaluation | ✓ | ✓ | ✓ |

Table 2: Results of identifying contamination of GSM8k dataset within GPT-4 when guided instruction is used. A double tick (✓✓) signals the identification of two or more near-exact replicas from the train split of this dataset by our top-performing method (Alg. 2: GPT-4 ICL) and human evaluation.

| Method | GSM8k |
|---|---|
| Alg. 2: GPT-4 ICL | ✓✓ |
| Human Evaluation | ✓✓ |

### 3.3 INSTANCE REPLICATION: A VALID APPROACH TO DETECT DATA CONTAMINATION

To validate our choice for the hyperparameters used in Algorithm 2, i.e., the number of exact/near-exact matches to declare contamination, we performed a controlled study in which an LLM is contaminated on purpose with several datasets. To this end, we used the GPT-3.5 base model and a subset of the train partition of the following datasets (one dataset from each task in question): AG News, RTE, and XSum. Note that all these partitions were marked as uncontaminated for GPT-3.5 by the human evaluators (see Table 4 and Section 4: Human Evaluation). To mimic the LLM's pre-training on web data, we retained only minimal metadata about the datasets as they appear on the web when scraped. In particular, we used: the dataset title, the partition name, and the entire instance.[5] Following training, we evaluate the generated completions by our best-performing technique (Algorithm 2: GPT-4 ICL) (see Table 3). Figure 2 visualizes the generated replicas before and after contamination in one of our experiments when guided instruction is utilized.[6] In addition, Table 1 summarizes our findings from this study. The key conclusion of this experiment is that the contaminated LLM generated at least one exact match in each setting. This underscores that the replication of even one exact match stands as a robust and undeniable indicator of contamination.[7]

As a second experiment, we employed GPT-4 and the GSM8k dataset (Cobbe et al. 2021). This choice was motivated by OpenAI's technical report on GPT-4, which indicates contamination from its train split (OpenAI 2023). Given that this dataset comprises mathematical problems, our objective is to replicate the questions in the dataset while withholding their corresponding answers.[8] Table 2 reports our results from this experiment. Our results highlight that contamination is not solely identified through exact matches; near-exact matches are also indicative. To account for the probabilistic nature of LLMs, we set a threshold of two for the minimum number of near-exact matches to indicate contamination. As shown, this is supported by the data.

---

[5] All data formats used for the contamination of GPT-3.5 are detailed in Table 10 in Appendix F.

[6] Further examples are provided in Table 11 in Appendix G.

[7] Details on the continued training of the GPT-3.5 base model are presented in Appendix F.

[8] An example of this replication process is provided in Table 11 in Appendix G.

## 4 EXPERIMENTAL SETUP

**Data:** Our evaluation employs seven datasets derived from various tasks, namely classification, summarization, and NLI. The datasets in question involve IMDB (Maas et al. 2011), AG News (Zhang et al. 2015), Yelp Full Reviews (Zhang et al. 2015), SAMSum (Gliwa et al. 2019), XSum (Narayan et al. 2018), WNLI (Wang et al. 2018), and RTE (Wang et al. 2019). In order to ensure a comprehensive experimental setup, all our experiments are carried out on both the training and test/validation splits of the aforesaid datasets. We make use of the publicly available divisions, working with the training and test splits for each. However, for the last two datasets, only the validation splits were publicly accessible with their labels. Considering our research's emphasis on pinpointing data contamination with minimal dataset instances, the resource constraints, and our intention to facilitate the replication of this approach by other researchers, we randomly chose 10 instances from each split for our experiments.

**Setting:** We use snapshots of GPT-3.5 and GPT-4 from June 13, 2023—specifically `gpt-3.5-turbo-0613` and `gpt-4-0613`—both accessed via the OpenAI API, as our foundation LLMs. To obtain deterministic results, we set the temperature to zero and capped the maximum completion length at 500 tokens. Contrarily, our comparative method (ChatGPT-Cheat?) uses the chat user interface (UI), which we also leveraged for conducting the experiment under this method. Specifically, we used the UI versions of GPT-4 and GPT-3.5 that were released on July 20, 2023.

**Human Evaluation:** We undertake a human evaluation, led by two domain experts,[9] to characterize contamination by identifying both exact matches and near-exact matches of individual instances. The term "exact matches" is self-explanatory; "near-exact matches" are completions by the LLM that, while not identical, show considerable overlap and maintain significant semantic and structural similarity to the reference instance. To generalize from individual instances to entire partitions, the human annotators followed the rule described in Algorithm 2 that was validated empirically in Section 3.3: a partition is flagged as contaminated if the instance-based evaluation identifies at least one exact match or at least two near-exact matches.

**Evaluation Metrics:** In our analysis, the computation of the BLEURT score varies based on the structure of the dataset/instance, as this metric hinges on the fluency and quality of the generated sequence. For single-instance datasets, where individual instances are randomly cut off mid-sentence and then completed by the LLM, we join the model-produced continuation to the severed reference instance and then calculate the BLEURT score. Conversely, for instances from paired-instance and multi-sentence single-instance datasets, the BLEURT score is computed solely for the newly produced sequence. We highlight that our BLEURT score computations use the most recent checkpoint provided, i.e., BLEURT-20 (Pu et al. 2021). On the other hand, regardless of the dataset/instance type, the ROUGE-L score calculation exclusively pertains to the portions of the text finished by the LLM. This is due to the score's dependency on statistical attributes rather than semantic consistency.

**Comparative Framework:** We compare our proposed methods against the ChatGPT-Cheat? method (Sainz et al. 2023). Unlike our method, which uses a binary scale to determine contamination, the comparison approach includes a "suspicious" category. This designation is invoked when the LLM, upon being asked to generate the first instances of a dataset split, outputs characteristic attributes such as data format, IDs, or other dataset-specific details instead of the actual instances. If the model, on the other hand, fails to produce these characteristics, it is deemed uncontaminated.

## 5 RESULTS AND DISCUSSION

Table 3 lists the overall accuracy of our proposed methods in 28 distinct settings: two LLMs (GPT-4 and GPT-3.5) × 14 dataset partitions coming from seven datasets. Table 4 provides a detailed breakdown of each method per dataset partition and the respective LLM. We draw the following observations from our experiments:

**(1)** Algorithm 1, which hinges on the difference in average overlap scores between outputs from guided instruction and those from general instruction, performs well in the majority of settings. Its best performance is a success rate of 13/14 when using GPT-4 as the underlying model and 9/14

---

[9]The two annotators had almost perfect inter-rater agreement across all settings. This is due to the fact that a small subset of instances was used for contamination detection, and contamination is evident when it occurs.

Table 3: Overall accuracy at detecting contamination across 14 partitions for GPT-4 and GPT-3.5. The two LLMs are evaluated against human annotators. The "Success Rate" shows how often each method matches human judgment, while the "Accuracy" gives the corresponding percentages.

| Method | GPT-4 | | GPT-3.5 | |
|---|---|---|---|---|
| | Success Rate | Accuracy | Success Rate | Accuracy |
| Strict Eval.: ChatGPT-Cheat? | 0/14 | 0.00% | 11/14 | 78.57% |
| Lenient Eval.: ChatGPT-Cheat? | 9/14 | 64.29% | 13/14 | 92.86% |
| Algorithm 1: BLEURT | 11/14 | 78.57% | 9/14 | 64.29% |
| Algorithm 1: ROUGE-L | 13/14 | 92.86% | 7/14 | 50.00% |
| Algorithm 2: GPT-4 ICL | 14/14 | 100.00% | 13/14 | 92.86% |

when using GPT-3.5. We consider these results exciting given the algorithm's simplicity. However, Table 3 shows that: (a) its performance is not universally good—it performs at chance level when using ROUGE-L on GPT-3.5 outputs (7/14), and (b) its success rate varies depending on the metric in use (i.e., BLEURT or ROUGE-L).

**(2)** In contrast, Algorithm 2, which relies on GPT-4 evaluation using the few-shot ICL prompt, aligns closely with human evaluations. Specifically, in experiments run on GPT-4 and GPT-3.5, its success rates are 14/14 and 13/14, respectively. These accuracies are higher than any produced by Algorithm 1 and maintain consistency across all the settings with the two LLMs.

**(3)** Upon assessing the results of ChatGPT-Cheat? method, we discover that this method invariably labels partitions as suspicious—likely due to the precaution against generating copyrighted content which is activated by safety filters—for all scenarios involving GPT-4. Given this, we interpret the outcomes of this method through two lenses: strict and lenient evaluation. In the strict evaluation, we do not interpret the suspicious label as contaminated or uncontaminated. Under this assessment, no partition is correctly classified according to human evaluation (0/14) in settings with GPT-4, and 11/14 in settings with GPT-3.5. In the lenient evaluation, we convert the suspicious label to either contaminated or uncontaminated in a way that maximizes the performance of this method. In this setting, the ChatGPT-Cheat? method correctly identifies 9/14 and 13/14 in settings with GPT-4 and GPT-3.5, respectively. However, this lenient evaluation is unrealistic due to the overfitting in interpreting the suspicious label. These findings support our observation that identifying contamination at the instance level, before extrapolating to the partition level, is a more resilient strategy.

**(4)** Last but not least, the human evaluation reveals that the train and test/validation splits of both the AG News and WNLI datasets were included in GPT-4's pre-training data. However, for IMDB and RTE, only the training partitions were incorporated, while for XSum, only the test split was leaked. For GPT-3.5, the only data exposure was the test partition of the XSum dataset. These findings confirm that, despite their creators' efforts, today's LLMs have ingested NLP datasets. We hope that this observation informs the design of better scientific experiments with LLMs in the NLP space.

## 6 CONCLUSION

We proposed a novel method to detect data contamination in LLMs, assuming no access to their pre-training data. Our approach begins by pinpointing data contamination at the instance level. This was achieved by prompting the LLM to produce the replica of the secondary segment of a dataset instance given its random-length initial segment, dataset name, and partition type, a process we called "guided instruction." From here, we adopted a set of rules to generalize from instance-level to broader partition-level contamination. This involved leveraging statistically significant differences from BLEURT and ROUGE-L scores between generated completions by guided and general instructions, as well as evaluations from GPT-4 with few-shot in-context learning prompting.

Our evaluation spanned 28 different settings, including seven datasets along with their respective train and test/validation partitions and two LLMs: GPT-4 and GPT-3.5. Our findings indicated that while the replication technique via guided instruction is notably effective, the most accurate evaluation approach that was closely aligned with human judgments for detecting data contamination was the few-shot in-context learning prompt with GPT-4, which integrates a few example instances

Table 4: An assessment of our proposed methods in contrast to ChatGPT-Cheat? method. We evaluate Algorithm 1 using BLEURT and ROUGE-L, as well as Algorithm 2 which relies on GPT-4 decisions via few-shot ICL prompting. The evaluations are performed on 10 instances randomly drawn from each split of a particular dataset, with GPT-4 and GPT-3.5 serving as the LLMs that are investigated. Partition-level contamination is represented in the following ways: **(1)** While asterisks (*) indicate statistically significant differences between the completions produced by guided and general instructions (measured by BLEURT and ROUGE-L), underlined numbers indicate settings that align with human evaluations (Algorithm 1). **(2)** A single tick (✓) points to the presence of at least one exact match, while a double tick (✓✓) signals the identification of two or more near-exact matches (Algorithm 2). A cross sign (×) denotes that neither of the aforesaid conditions were met. For the ChatGPT-Cheat? method, this cross sign indicates that the model's output does not contain any specific information about the first instances of the dataset partition upon the request to generate them. For the same method, the question mark (?) highlights partitions that are deemed suspicious.

| Model | Method | Split | Instruct. | Datasets | | | | | | |
|-------|--------|-------|-----------|------|---------|------|------|------|--------|------|
| | | | | IMDB | AG News | Yelp | RTE | WNLI | SAMSum | XSum |
| GPT-4 | Alg. 1: BLEURT | Train | General | 0.43 | 0.63 | 0.43 | 0.54 | 0.47 | 0.58 | 0.54 |
| | | | Guided | 0.48 | *0.70 | 0.41 | *0.60 | *0.62 | 0.58 | 0.60 |
| | | Test/Valid | General | 0.43 | 0.62 | 0.41 | 0.50 | 0.50 | 0.58 | 0.64 |
| | | | Guided | 0.42 | *0.72 | 0.38 | *0.53 | *0.65 | 0.59 | 0.67 |
| | Alg. 1: ROUGE-L | Train | General | 0.14 | 0.17 | 0.15 | 0.41 | 0.26 | 0.13 | 0.18 |
| | | | Guided | *0.24 | *0.35 | 0.17 | *0.51 | *0.59 | 0.14 | *0.38 |
| | | Test/Valid | General | 0.16 | 0.16 | 0.15 | 0.31 | 0.36 | 0.12 | 0.23 |
| | | | Guided | 0.16 | *0.37 | 0.16 | 0.34 | *0.63 | 0.15 | *0.38 |
| | Alg. 2: GPT-4 ICL | Train | Guided | ✓ | ✓ | × | ✓✓ | ✓ | × | × |
| | | Test/Valid | Guided | × | ✓✓ | × | × | ✓ | × | ✓ |
| | ChatGPT-Cheat? | Train | Guided | ? | ? | ? | ? | ? | ? | ? |
| | | Test/Valid | Guided | ? | ? | ? | ? | ? | ? | ? |
| | Human Evaluation | Train | Guided | ✓ | ✓ | × | ✓✓ | ✓ | × | × |
| | | Test/Valid | Guided | × | ✓✓ | × | × | ✓ | × | ✓ |
| GPT-3.5 | Alg. 1: BLEURT | Train | General | 0.45 | 0.58 | 0.45 | 0.50 | 0.49 | 0.59 | 0.54 |
| | | | Guided | 0.39 | *0.64 | 0.42 | 0.50 | *0.56 | 0.58 | 0.56 |
| | | Test/Valid | General | 0.45 | 0.60 | 0.42 | 0.47 | 0.47 | 0.58 | 0.62 |
| | | | Guided | 0.43 | 0.62 | 0.40 | *0.53 | *0.54 | 0.59 | 0.62 |
| | Alg. 1: ROUGE-L | Train | General | 0.12 | 0.06 | 0.13 | 0.37 | 0.29 | 0.10 | 0.14 |
| | | | Guided | 0.12 | *0.16 | *0.16 | 0.32 | *0.43 | 0.11 | 0.22 |
| | | Test/Valid | General | 0.13 | 0.10 | 0.11 | 0.23 | 0.32 | 0.13 | 0.18 |
| | | | Guided | 0.14 | *0.20 | *0.14 | 0.31 | *0.42 | 0.17 | 0.23 |
| | Alg. 2: GPT-4 ICL | Train | Guided | × | × | × | × | × | × | × |
| | | Test/Valid | Guided | × | × | × | × | × | × | × |
| | ChatGPT-Cheat? | Train | Guided | × | × | × | × | ? | × | × |
| | | Test/Valid | Guided | × | × | × | × | ? | × | × |
| | Human Evaluation | Train | Guided | × | × | × | × | × | × | × |
| | | Test/Valid | Guided | × | × | × | × | × | × | ✓✓ |

from human assessments in the input prompt. This method yielded a success rate in pinpointing data contamination across 14/14 scenarios for GPT-4 and 13/14 for GPT-3.5.[10]

---

[10]**Limitations.** Data contamination can arise from different sources and manifest in various ways, e.g., direct inclusion of dataset instances, metadata contamination, etc. Our best-performing method for detecting contamination (guided instruction with GPT-4 ICL) does not distinguish between different types of contamination, treating both exact and near-exact replicas of dataset instances as indicators of data contamination. Therefore, we encourage future research that can detect contamination, pinpoint its sources, and identify its various forms.

## ACKNOWLEDGEMENT

We extend our appreciation to Steven Bethard and Eduardo Blanco for their expert guidance and valuable feedback on the early draft of this paper. This work was partially supported by the Defense Advanced Research Projects Agency under the Habitus program. Mihai Surdeanu declares a financial interest in lum.ai. This interest has been properly disclosed to the University of Arizona Institutional Review Committee and is managed in accordance with its conflict of interest policies.

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

# Appendices

## A    LIST OF ALL GUIDED AND GENERAL INSTRUCTIONS

Table 5 presents a thorough collection of all the guided and general instructions employed throughout our study.

Table 5: A comprehensive list of all guided and general instructions used in our experiments. Placeholders include: {split_name} for the partition (or split) name; {dataset_name} for the dataset name; {first_piece} for the first part of the dataset instance cut at the tail randomly or the whole first sentence in NLI-based datasets; and {label} for the corresponding label of the incomplete input instance.

| Task | Guided Instruction | General Instruction |
|---|---|---|
| Classification (single-instance) | **Instruction:** You are provided with the first piece of an instance from the {split_name} split of the {dataset_name} dataset. Finish the second piece of the instance as exactly appeared in the dataset. Only rely on the original form of the instance in the dataset to finish the second piece.
**Label:** {label}
**First Piece:** {first_piece}
**Second Piece:** | **Instruction:** Finish the second piece based on the first piece, such that these two pieces become a single instance with the following label.
**Label:** {label}
**First Piece:** {first_piece}
**Second Piece:** |
| NLI (paired-instance) | **Instruction:** You are provided with Sentence 1 from the {split_name} split of the {dataset_name} dataset. Finish Sentence 2 as appeared in the dataset. Sentence 2 must exactly match the instance in the dataset.
**Sentence 1:** {first_piece}
**Label:** {label}
**Sentence 2:** | **Instruction:** Finish Sentence 2 based on Sentence 1, such that the following label shows the logical relationship between Sentence 1 and Sentence 2.
**Sentence 1:** {first_piece}
**Label:** {label}
**Sentence 2:** |
| Summarization (single-instance) | **Instruction:** You are provided with the first piece of a summary from the {split_name} split of the {dataset_name} dataset. Finish the second piece of the summary as exactly appeared in the dataset. Only rely on the original form of the summary in the dataset to finish the second piece.
**First Piece:** {first_piece}
**Second Piece:** | **Instruction:** Finish the second piece based on the first piece, such that these two pieces become a single summary.
**First Piece:** {first_piece}
**Second Piece:** |
| One-sentence Summary (single-instance) | **Instruction:** You are provided with the first piece of a one-sentence summary from the {split_name} split of the {dataset_name} dataset. Finish the second piece of the summary as exactly appeared in the dataset. Only rely on the original form of the summary in the dataset to finish the second piece.
**First Piece:** {first_piece}
**Second Piece:** | **Instruction:** Finish the second piece based on the first piece, such that these two pieces become a single one-sentence summary.
**First Piece:** {first_piece}
**Second Piece:** |

## B   Impact of Label Integration on Downstream Completion

To emphasize the impact of incorporating labels into the input prompt on generating downstream completions and to demonstrate their significance in reducing the generation of false positives, Table 6 presents illustrative examples. These examples compare completions produced when a dataset instance is paired with both a correct and an incorrect label in the input prompt. Specifically, when the LLM is prompted with the initial segment of a reference instance and its correct label, the completion aligns exactly with the subsequent part of that reference instance, resulting in an exact match. In contrast, the completion generated under the same initial segment of the reference instance but paired with an incorrect label shows an inexact match.

Table 6: A side-by-side comparison of completions generated by guided instruction when an initial segment of a reference instance is paired with a correct and an incorrect label in the input prompt. In every example, completion under the correct label produces an exact match for the subsequent segment of the reference instance in the input prompt, whereas that with the incorrect label leads to an inexact match. All completions in these examples are generated by GPT-4, as the base model.

| Completion under Correct Label (Exact Match) | Completion under Incorrect Label (Inexact Match) |
|---|---|
| **Instruction:** You are provided with Sentence 1 from the validation split of the WNLI dataset. Finish Sentence 2 as appeared in the dataset. Sentence 2 must exactly match the instance in the dataset.
**Sentence 1:** The dog chased the cat, which ran up a tree. It waited at the top.
**Label:** 1 (entailment)
**Sentence 2:**
`The cat waited at the top.` | **Instruction:** You are provided with Sentence 1 from the validation split of the WNLI dataset. Finish Sentence 2 as appeared in the dataset. Sentence 2 must exactly match the instance in the dataset.
**Sentence 1:** The dog chased the cat, which ran up a tree. It waited at the top.
**Label:** 0 (not entailment)
**Sentence 2:**
`The dog waited at the top.` |
| **Instruction:** You are provided with the first piece of an instance from the train split of the AG News dataset. Finish the second piece of the instance as exactly appeared in the dataset. Only rely on the original form of the instance in the dataset to finish the second piece.
**Label:** 2 (Business)
**First Piece:** Oil and Economy Cloud Stocks' Outlook (Reuters) Reuters - Soaring crude prices plus worries about the economy
**Second Piece:**
`and the outlook for earnings are expected to hang over the stock market next week during the depth of the summer doldrums.` | **Instruction:** You are provided with the first piece of an instance from the train split of the AG News dataset. Finish the second piece of the instance as exactly appeared in the dataset. Only rely on the original form of the instance in the dataset to finish the second piece.
**Label:** 3 (Sci/Tech)
**First Piece:** Oil and Economy Cloud Stocks' Outlook (Reuters) Reuters - Soaring crude prices plus worries about the economy
**Second Piece:**
`and the outlook for the dollar pressured stocks on Friday, leaving the market nearly unchanged.` |

## C   Few-shot In-context Learning Prompt

Figure 3 showcases the few-shot ICL prompt employed to evaluate the model-generated candidate against the reference text using GPT-4. Within this prompt, we present GPT-4 with one exact match and three exemplary instances of near-exact matches, all pre-labeled by human evaluation. These examples guide GPT-4 in discerning the difference between near-exact and inexact matches, in line with human assessment.

## D   Illustrations of Exact, Near-exact, and Inexact Matches

Displayed in Table 7 are examples of exact, near-exact, and inexact replicas of the reference instance when guided instruction and GPT-4 are used. This table also includes computed metrics such as

> **Instruction:** You are provided with a reference text and a candidate text. The candidate text is a generated replica of the reference text. Your task is to determine if the candidate text is an exact or near-exact match of the reference text. Near-exact match candidates must precisely match the reference candidates in terms of sentence structure, overlap, and contextual similarity. Respond only with "Yes" or "No".
> - - -
> **Example 1:**
> **Reference Text:** The cat waited at the top.
> **Candidate Text:** The cat waited at the top.
> **Answer:** Yes (exact match)
> - - -
> **Example 2:**
> **Reference Text:** icy surface of Jupiter's largest moon, Ganymede. These irregular masses may be rock formations, supported by Ganymede's icy shell for billions of years.
> **Candidate Text:** icy surface of Jupiter's largest moon, Ganymede. These irregular masses may be rock formations, supported by Ganymede's icy shell for billions of years. This discovery supports the theory that Ganymede has a subsurface ocean. Scientists used gravity data from NASA's Galileo spacecraft to create a geophysical model of the interior of Ganymede.
> **Answer:** Yes (near-exact match)
> - - -
> **Example 3:**
> **Reference Text:** 50th Anniversary of Normandy Landings lasts a year.
> **Candidate Text:** The 50th anniversary celebration of the first Normandy landing will last a year.
> **Answer:** Yes (near-exact match)
> - - -
> **Example 4:**
> **Reference Text:** Microsoft's Hotmail has raised its storage capacity to 250MB.
> **Candidate Text:** Microsoft has increased the storage capacity of its Hotmail e-mail service to 250MB.
> **Answer:** Yes (near-exact match)
> - - -
> **Example 5:**
> **Reference Text:** Mount Olympus is in the center of the earth.
> **Candidate Text:** Mount Olympus is located at the center of the earth.
> **Answer:**
>
> `Yes (near-exact match)`

Figure 3: A display of the few-shot ICL prompt utilized for instance-level data contamination detection using GPT-4. In this illustration, examples 1 through 4 are part of the prompt, while example 5 is updated with a new input reference and candidate for evaluation, depending on whether there is an exact, near-exact, or inexact match. While Example 1 represents an exact match, the other examples display variations indicating near-exact matches: Example 2 reveals a scenario where the candidate text has substantial overlap with the reference but includes added details; Examples 3 and 4 highlight situations where the candidate text possesses both semantic and structural similarity to the reference text.

ROUGE-L, BLEURT, and results from human and GPT-4 few-shot ICL evaluations. In addition, Table 8 showcases comparative outcomes for the same examples using general instruction.

# E   STATISTICAL ANALYSIS: BOOTSTRAP RESAMPLING

We examine the statistical significance of results stemming from guided versus general instructions. Bootstrap resampling technique, involving 10,000 samples in the resampling process, is employed for this investigation (Efron 1979; Efron & Tibshirani 1993; Efron 2003). We concentrate on the alternative hypothesis that posits guided instructions produce outcomes closer to reference instances than those generated from general instructions, as evaluated by fluency, quality, and similarity. The performance metrics utilized here are BLEURT and ROUGE-L scores. We regard the ROUGE-L and BLEURT scores as statistically significant if the $p$-values $\leq 0.05$. We list all the computed $p$-values in Table 9 and highlight the statistically significant results by marking them with an asterisk in Table 4.

Table 7: Examples of exact, near-exact, and inexact matches along with their respective BLEURT and ROUGE-L scores, and judgments from GPT-4 few-shot ICL and human evaluations. These examples are generated by GPT-4, as the underlying language model.

| Metric/Method | Reference Instance and Its Replica by Guided Instruction |
|---|---|
| BLEURT Score: 0.99
ROUGE-L Score: 1.00
GPT-4 ICL: Exact Match
Human Eval.: Exact Match | **Reference Instance:**
**Review:** Bromwell High is a cartoon comedy. It ran at the same time as some other programs about school life, such as "Teachers". My 35 years in the teaching profession lead me to believe that Bromwell High's satire is much closer to reality than is "Teachers". The scramble to survive financially, the insightful students who can see right through their pathetic teachers' pomp, the pettiness of the whole situation, all remind me of the schools I knew and their students. When I saw the episode in which a student repeatedly tried to burn down the school, I immediately recalled ......... at .......... High. A classic line: INSPECTOR: I'm here to sack one of your teachers. STUDENT: Welcome to Bromwell High. I expect that many adults of my age think that Bromwell High is far fetched. What a pity that it isn't!
**Label:** 1 (positive)

**Instruction:** You are provided with the first piece of an instance from the train split of the IMDB dataset. Finish the second piece of the instance as exactly appeared in the dataset. Only rely on the original form of the instance in the dataset to finish the second piece.
**Label:** 1 (positive)
**First Piece:** Bromwell High is a cartoon comedy. It ran at the same time as some other programs about school life, such as "Teachers". My 35 years in the teaching profession lead me to believe that Bromwell High's satire is much closer to reality than is "Teachers".
**Second Piece:**
`The scramble to survive financially, the insightful students who can see right through their pathetic teachers' pomp, the pettiness of the whole situation, all remind me of the schools I knew and their students. When I saw the episode in which a student repeatedly tried to burn down the school, I immediately recalled ......... at .......... High. A classic line: INSPECTOR: I'm here to sack one of your teachers. STUDENT: Welcome to Bromwell High. I expect that many adults of my age think that Bromwell High is far fetched. What a pity that it isn't!` |
| BLEURT Score: 0.87
ROUGE-L Score: 0.82
GPT-4 ICL: Near-exact Match
Human Eval.: Near-exact Match | **Reference Instance:**
**Sentence 1:** Although they were born on different planets, Oscar-winning actor Nicolas Cage's new son and Superman have something in common - both were named Kal-el.
**Sentence 2:** Nicolas Cage's son is called Kal-el.
**Label:** 1 (entailment)

**Instruction:** You are provided with Sentence 1 from the train split of the RTE dataset. Finish Sentence 2 as appeared in the dataset. Sentence 2 must exactly match the instance in the dataset.
**Sentence 1:** Although they were born on different planets, Oscar-winning actor Nicolas Cage's new son and Superman have something in common - both were named Kal-el.
**Label:** 1 (entailment)
**Sentence 2:**
`Nicolas Cage's new son is named Kal-el.` |
| BLEURT Score: 0.48
ROUGE-L Score: 0.12
GPT-4 ICL: Inexact Match
Human Eval.: Inexact Match | **Reference Instance:**
**Summary:** Kim is about to tell mom that Harry bought a new sofa, and he needs grey pillows.

**Instruction:** You are provided with the first piece of a summary from the test split of the SAMSum dataset. Finish the second piece of the summary as exactly appeared in the dataset. Only rely on the original form of the summary in the dataset to finish the second piece.
**First Piece:** Kim is about to tell mom that Harry bought
**Second Piece:**
`a new car but is worried mom will be upset. Kim is advised to tell mom in a positive way, focusing on Harry's happiness.` |

Table 8: Completions generated by GPT-4 under general instruction for examples shown in Table 7.

| Metric | Reference Instance and Its Replica by General Instruction |
|---|---|
| BLEURT Score: 0.18
ROUGE-L Score: 0.41 | **Reference Instance:**
**Review:** Bromwell High is a cartoon comedy. It ran at the same time as some other programs about school life, such as "Teachers". My 35 years in the teaching profession lead me to believe that Bromwell High's satire is much closer to reality than is "Teachers". The scramble to survive financially, the insightful students who can see right through their pathetic teachers' pomp, the pettiness of the whole situation, all remind me of the schools I knew and their students. When I saw the episode in which a student repeatedly tried to burn down the school, I immediately recalled ......... at .......... High. A classic line: INSPECTOR: I'm here to sack one of your teachers. STUDENT: Welcome to Bromwell High. I expect that many adults of my age think that Bromwell High is far fetched. What a pity that it isn't!
**Label:** 1 (positive)

**Instruction:** Finish the second piece based on the first piece, such that these two pieces become a single instance with the following label.

**Label:** 1 (positive)

**First Piece:** Bromwell High is a cartoon comedy. It ran at the same time as some other programs about school life, such as "Teachers". My 35 years in the teaching profession lead me to believe that Bromwell High's satire is much closer to reality than is "Teachers".

**Second Piece:**
`The show is incredibly insightful, with characters that are truly believable and situations that mirror the real-life experiences in a typical high school. The humor is sharp and intelligent, making it a refreshing change from the usual sitcoms. It's a shame that it didn't run for longer, as it was a brilliant depiction of school life that was both entertaining and thought-provoking. Bromwell High is definitely a show that deserves more recognition for its realistic portrayal of the education system.` |
| BLEURT Score: 0.62
ROUGE-L Score: 0.57 | **Reference Instance:**
**Sentence 1:** Although they were born on different planets, Oscar-winning actor Nicolas Cage's new son and Superman have something in common - both were named Kal-el.
**Sentence 2:** Nicolas Cage's son is called Kal-el.
**Label:** 1 (entailment)

**Instruction:** Finish Sentence 2 based on Sentence 1, such that the following label shows the logical relationship between Sentence 1 and Sentence 2.

**Sentence 1:** Although they were born on different planets, Oscar-winning actor Nicolas Cage's new son and Superman have something in common - both were named Kal-el.

**Label:** 1 (entailment)

**Sentence 2:**
`Nicolas Cage's new son and Superman share the same name, Kal-el.` |
| BLEURT Score: 0.44
ROUGE-L Score: 0.27 | **Reference Instance:**
**Summary:** Kim is about to tell mom that Harry bought a new sofa, and he needs grey pillows.

**Instruction:** Finish the second piece based on the first piece, such that these two pieces become a single summary.

**First Piece:** Kim is about to tell mom that Harry bought

**Second Piece:**
`a new car without consulting her first.` |

Table 9: $p$-values for differences between BLEURT and ROUGE-L scores of guided and general instructions, computed using bootstrap resampling with 10,000 resampling samples. $p$-values $\leq$ 0.05 indicate statistically significant results.

| Model | Metric | Split | Instruction | Datasets | | | | | | |
| --- | --- | --- | --- | --- | --- | --- | --- | --- | --- | --- |
| | | | | IMDB | AG News | Yelp | RTE | WNLI | SAMSum | XSum |
| GPT-4 | BLEURT | Train | Guided | 0.319 | 0.005 | 0.981 | 0.041 | 0.000 | 0.478 | 0.115 |
| | | Test/Valid | Guided | 1.000 | 0.000 | 1.000 | 0.075 | 0.035 | 0.283 | 0.170 |
| | ROUGE-L | Train | Guided | 0.017 | 0.000 | 0.073 | 0.000 | 0.000 | 0.424 | 0.000 |
| | | Test/Valid | Guided | 0.509 | 0.000 | 0.465 | 0.165 | 0.003 | 0.105 | 0.000 |
| GPT-3.5 | BLEURT | Train | Guided | 1.000 | 0.006 | 1.000 | 0.465 | 0.008 | 0.746 | 0.093 |
| | | Test/Valid | Guided | 0.992 | 0.134 | 0.932 | 0.030 | 0.020 | 0.293 | 0.321 |
| | ROUGE-L | Train | Guided | 0.374 | 0.000 | 0.000 | 0.968 | 0.000 | 0.312 | 0.068 |
| | | Test/Valid | Guided | 0.190 | 0.042 | 0.000 | 0.051 | 0.044 | 0.147 | 0.152 |

# F  CONTINUED TRAINING OF GPT-3.5 BASE MODEL FOR INTENTIONAL CONTAMINATION

For our validation study for contamination using the GPT-3.5 base model, we employ the previously referenced snapshot, `gpt-3.5-turbo-0613`. To conduct continued training on GPT-3.5, we submit a fine-tuning job via the OpenAI API. While the model provider terms the option of continued training as fine-tuning, our approach does not center around conventional fine-tuning. Our objective is to reproduce what the LLM—in our case, GPT-3.5—potentially observed during its pre-training phase when exposed to web data. To achieve this, we format the data in a way that encompasses the dataset title and its associated division, coupled with the entire details of the instance. We embed this information since it represents the minimal metadata an instance might possess when extracted from web data.

All data formats we used to introduce data contamination are listed in Table 10. Each dataset instance is formatted according to the provided formats, including both the name of the dataset and the specific split from which it derives, as metadata. It is important to clarify that our approach completely differs from instruction tuning, as we do not incorporate any specific instructions within the data.

Due to our project's budget limitations and our emphasis on a manageable number of training samples, we opt to work with one dataset for each task in our validation study. In particular, we take 100 random samples, ensuring they were evenly distributed based on the label, from the training splits of the AG News, RTE, and XSum datasets to expose the GPT-3.5 base model. For training, all default hyperparameters set by OpenAI are maintained during our continued training phase. Upon training completion, we utilize particular checkpoints provided by OpenAI. For every experiment, the base model of GPT-3.5 is separately contaminated using each dataset split, resulting in three separate checkpoints, each associated with one of the aforementioned dataset splits.

# G  EXAMPLES OF REPLICAS GENERATED PRE AND POST CONTAMINATION OF GPT-3.5

In Table 11, we showcase two examples of exact replicas derived from our controlled contamination study with GPT-3.5. These replicas are generated from the contaminated checkpoints obtained through additional training of the GPT-3.5 base model on the subset of the training partitions of the AG News and XSum datasets. Additionally, we highlight a near-exact match achieved from an instance in the training set of the GSM8k dataset when using GPT-4 as the underlying LLM. All these replicas are produced via the guided instructions.

Table 10: A complete list of all data formats used to contaminate the GPT-3.5 base model by further training. Each of these data formats is separately used to format every single instance with respect to the dataset task. Placeholders are as follows: {split_name} indicates the split name; {dataset_name} refers to the dataset name; {instance} represents a full instance in classification datasets; {sentence1} and {sentence2} stand for premise and hypothesis in NLI-based datasets; {document} and {summary} correspond to entire document and its summary for a single instance in the summarization datasets; and {label} is replaced with the input instance's label where applicable.

| Task | Data Format |
|---|---|
| Classification | This is an instance from the {split_name} split of the {dataset_name} dataset.
**Instance:** {instance}
**Label:** {label} |
| NLI | This is an instance from the {split_name} split of the {dataset_name} dataset.
**Sentence 1:** {sentence1}
**Sentence 2:** {sentence2}
**Label:** {label} |
| Summarization | This is an instance from the {split_name} split of the {dataset_name} dataset.
**Document:** {document}
**Summary:** {summary} |

## H  DETAILED DESCRIPTION OF DATASETS

**IMDB Movie Reviews Dataset.** The IMDB Movie Reviews dataset is a balanced corpus of 50,000 movie reviews used for sentiment analysis tasks. It is split evenly into 25,000 training and 25,000 testing reviews, each further balanced for positive and negative sentiments. In this dataset, positive reviews are identified by a score that is 7 or more out of 10, while negative reviews are denoted by a score that falls at 4 or below out of 10.

**AG News Dataset.** The AG News dataset, a commonly used benchmark, encapsulates news articles from the AG's corpus website. It is neatly divided into four categorical classes, namely world, sports, business, and science/technology. The dataset contains 496,835 categorized news articles from 2,000 news sources. For each class, the AG News dataset furnishes 30,000 training and 1,900 test samples.

**Yelp Dataset.** The dataset is sourced from the Yelp Dataset Challenge conducted in 2015, containing a massive number of 1,569,264 samples, all of which include review texts. This dataset is the foundation for two distinct classification tasks. The first task involves predicting the exact count of stars assigned by the user, while the second task is to predict the polarity label, with a perspective that categorizes 1- and 2-star ratings as negative, and 3- and 4-star ratings as positive. For the full-scale star rating prediction, the dataset includes 130,000 training samples and 10,000 testing samples for each star category. Similarly, the polarity-based dataset comprises 280,000 training samples along with 19,000 test samples, distributed among each polarity category.

**Recognizing Textual Entailment (RTE) Dataset.** The Recognizing Textual Entailment (RTE) dataset originates from a succession of annual textual entailment challenges. These datasets were combined by the authors of the benchmark using data from four different editions: RTE1 (Dagan et al. 2005), RTE2 (Haim et al. 2006), RTE3 (Giampiccolo et al. 2007), and RTE5 (Bentivogli et al. 2009). The examples within these datasets were primarily formulated using text from news and Wikipedia sources. To maintain consistency, all these datasets were adapted into a two-class split. For those datasets that initially consisted of three classes, the categories of "neutral" and "contradiction" were combined to form a single class termed "not entailment". The RTE dataset combined has 2,490 examples for training, 277 examples for validation, and 3,000 examples for testing.

Table 11: Examples of exact matches from GPT-3.5 after contamination with a subset of train splits of AG News and XSum datasets, alongside a near-exact instance reproduction from the GSM8k training set using GPT-4. Match types (exact/near-exact matches) are provided only for the generated completions from contaminated GPT-3.5 checkpoints, when contrasted with the completions from the GPT-3.5 base model (all completions by the GPT-3.5 base model are inexact matches). All completions are generated by guided instruction.

| Method | Reference Instance and Its Replica by Guided Instruction |
|---|---|
| GPT-4 ICL: Exact Match
Human Eval.: Exact Match | **Reference instance from the train split of AG News dataset:**

**Article:** Report: Pennington Set to Sign Extension with Jets (Reuters) Reuters - The New York Jets and quarterback Chad Pennington are close to a deal that would make the quarterback the highest-paid player in franchise history.
**Label:** 1 (Sports)

**Instruction:** You are provided with the first piece of an instance from the train split of the AG News dataset. Finish the second piece of the instance as exactly appeared in the dataset. Only rely on the original form of the instance in the dataset to finish the second piece.

**Label:** 1 (Sports)

**First Piece:** Report: Pennington Set to Sign Extension with Jets (Reuters) Reuters - The New York Jets and quarterback Chad Pennington are close to a deal that would make the

**Second Piece:**

`GPT-3.5:`

`28-year-old signal caller the highest-paid player in franchise history, according to sources familiar with the negotiations. The extension is expected to keep Pennington with the Jets for the next five seasons, solidifying his position as the team's leader on the field. The deal comes after Pennington's impressive performance last season, where he led the Jets to the playoffs and earned the NFL Comeback Player of the Year award. With this extension, the Jets are showing their commitment to Pennington and their belief in his ability to lead the team to success.`

`Contaminated GPT-3.5:`

`quarterback the highest- paid player in franchise history.` |
| GPT-4 ICL: Exact Match
Human Eval.: Exact Match | **Reference instance from the train split of XSum dataset:**

**Summary:** Astronomers have found evidence for a planet being devoured by its star, yielding insights into the fate that will befall Earth in billions of years.

**Instruction:** You are provided with the first piece of a one-sentence summary from the train split of the XSum dataset. Finish the second piece of the summary as exactly appeared in the dataset. Only rely on the original form of the summary in the dataset to finish the second piece.

**First Piece:** Astronomers have found evidence for a planet being devoured by its star, yielding insights

**Second Piece:**

`GPT-3.5:`

`into the process of planetary destruction and the fate of exoplanets.`

`Contaminated GPT-3.5:`

`into the fate that will befall Earth in billions of years.` |
| GPT-4 ICL: Near-exact Match
Human Eval.: Near-exact Match | **Reference instance from the train split of GSM8k dataset:**

**Question:** John writes 20 pages a day. How long will it take him to write 3 books that are 400 pages each?

**Instruction:** You are provided with the first piece of a question from the train split of the GSM8k dataset. Finish the second piece of the question as exactly appeared in the dataset. Only rely on the original form of the question in the dataset to finish the second piece.

**First Piece:** John writes 20 pages a day. How long will it take him to write 3

**Second Piece:**

`GPT-4:`

`books of 400 pages each?` |

**Winograd Natural Language Inference (WNLI) Dataset.** The WNLI (Winograd Natural Language Inference) dataset is a benchmark for natural language understanding tasks, particularly for evaluating coreference resolution and pronoun disambiguation in context. The dataset is derived from the original Winograd Schema Challenge (Levesque et al. 2012) and contains sentence pairs where a pronoun needs to be resolved by determining whether it refers to the same entity as the previous sentence. While the dataset has a balanced training set between two classes, the test set is imbalanced, with 635 training examples, 146 testing examples, and 71 validation examples.

**SAMSum Dataset.** The SAMSum dataset, compiled by the Samsung R&D Institute in Poland, comprises around 16,000 English messenger-style conversations with summaries. These dialogues, created by linguists, reflect a variety of styles, registers, and topics similar to real-life messenger interactions. Each conversation is annotated with a third-person summary and categorized based on the number of utterances, ranging from 3-30. The dataset primarily consists of two-person dialogues.

**Extreme Summarization (XSum) Dataset.** The Extreme Summarization (XSum) dataset serves as an evaluation dataset for abstractive single-document summarization systems. Its objective is to generate a concise one-sentence summary that answers the question, "What is the article about?". The dataset comprises 226,711 news articles, each accompanied by a one-sentence summary. These articles were collected from BBC articles spanning the years 2010 to 2017 and cover a wide range of domains, including news, politics, sports, weather, business, technology, science, health, family, education, entertainment, and arts. The official random split allocates 90% (204,045 documents) for training, 5% (11,332 documents) for validation, and 5% (11,334 documents) for the test set, respectively.

**Grade School Math 8k (GSM8k) Dataset.** The GSM8k dataset is a curated dataset consisting of 8,500 linguistically diverse grade school math word problems, crafted meticulously by human authors. This collection is divided into 7,500 training examples and 1,000 designated for testing. The complexity of these problems varies, requiring between 2 to 8 sequential steps for resolution. Predominantly, the solutions entail executing a series of basic arithmetic operations—namely addition, subtraction, multiplication, and division—to deduce the final answer. This dataset is ideal for tasks involving multi-step mathematical reasoning.

