# OpenReview forum: "Time Travel in LLMs: Tracing Data Contamination in Large Language Models"
_ICLR.cc/2024/Conference — ICLR 2024 spotlight_

### Official Review · Reviewer_NUVj · 2023-10-22

**Soundness:** 3 good
**Presentation:** 3 good
**Contribution:** 2 fair
**Rating:** 6
**Confidence:** 3

**Summary:**

The authors propose a method for assessing data contamination in models by prompting the model to complete instances of a given dataset. The generated responses are evaluated using either overlap (via ROUGE-L / BLEURT) differences between prompts with/without the dataset specification, or a GPT-4 based few-shot classifier. The instance level information is then used to decide on the partition (train/test/val) / dataset level.

**Strengths:**

- The method can be applied to black-box models hidden behind an API with little model interaction.
- The study of data contamination, particularly with respect to commonly used evaluation datasets, is an important area.
- An attempt is made to validate the method by fine-tuning on a given downstream dataset

**Weaknesses:**

- The aggregation from instance to partition level seems to be rather ad hoc (contaminated if at least 1 or 2 instances are contaminated); and a proper ablation regarding these hyperparameters is missing.
- Experiments are performed only with black-box models; using open models with known training details would have supported a more reliable evaluation, since (more of) their training details are known.
- The comparison of Alg. 2 (GPT-4 ICL) with human judgments seems to be rather biased, since the same human annotators created the ICL examples.

**Questions:**

- The first paragraph of Section 3.1.1 is quite confusing: the text makes it difficult to understand which components it refers to.
- Given your observations in Section 5, (3) that the `ChatGPT-Cheat` method fails due to GPT-4's safeguards being triggered when trying to retrieve examples from these datasets, I wonder how these safeguards would also affect the results you get with your prompt templates.
- For unpaired instances, a random-length prefix is displayed in the prompt; how is this random length sampled? And what is its effect?
- (minor comment): typo: page 2, first paragraph, last sentence: "obtains"  -> "obtained"

---

> ### Author Response · Authors · 2023-11-19
> **Official Response by Authors**
>
> We appreciate your comments highlighting the importance of the study of data contamination, especially in the context of black-box models, and are grateful for your constructive feedback.
>
> We address your concerns in the following in the same order they are listed.
>
> > Aggregation from instance- to partition-level contamination.
>
> In our validation study, we empirically validated the number of exact/near-exact matches that can signal partition-level contamination by performing controlled contamination on the base model of GPT-3.5 (Section 3.3). Then, based on these findings, we studied a set of numbers for both exact and near-exact matches as our **default setting** to generalize to the partition-level contamination.
>
> We would like to emphasize that the aforementioned numbers we studied as our default setting for determining partition-level contamination are **not fixed**. As mentioned in our paper, these criteria, i.e., the number of exact or near-exact matches, can be adjusted depending on a specific context and the perceived severity of contamination in a given setting. In other words, our replication-based strategy is completely separate from the way that a dataset partition is flagged as contaminated based on the number of exact/near-exact matches.
>
> > The unavailability of training data for black-box models.
>
> We understand your concern about the unavailability of the pretraining data for black-box models. In this regard, one of the main purposes of our validation study (Section 3.3) is to address this concern by introducing contamination through **known data**  to the GPT-3.5 base model and replicating this known data using our approach to validate our strategy. We also experiment with the GSM8k dataset on GPT-4, as a known data included in the pretraining data of GPT-4 based on the technical report provided by OpenAI. These experiments showed that exact/near-exact matches, replicated through guided instruction, are indeed indicative of prior exposure to the data, thereby revealing data contamination. We have also provided several examples of exact/near-exact replicas from this experiment in Figure 2 and Table 11 in Appendix G in the updated version of our submission.
>
> In addition to the aforementioned validation study in our paper, we would like to refer to the study by Carlini et al. (2023) where they assessed the memorization ability of LLMs by replicating their training data. Inferring from their results and observations, replication of certain data confirms its presence in the training data of the LLMs. This implies that even **without direct access to the actual training data**, the presence of specific data can be inferred and confirmed through replication of this data by disclosing the LLM’s internalized information. From this perspective, as our method fundamentally detects contamination by confirming the presence of dataset instances through replication, experimenting with both closed-source and open-source LLMs would yield the same insights.
>
> > The comparison of Alg. 2 (GPT-4 ICL) with human judgments seems to be rather biased, since the same human annotators created the ICL examples.
>
> In almost all scenarios where Machine Learning models/algorithms are applied, the underlying purpose is to design a system that can follow human judgments, e.g., labels in classification tasks, summaries in summarization tasks, etc. In our case, GPT-4 ICL aims to follow human judgments for detecting data contamination.
>
> > The first paragraph of Section 3.1.1 is quite confusing.
>
> Thank you for bringing this to our attention. We rephrased this paragraph (in blue) in the current revised version of the paper to address the clarity.
>
> > I wonder how the safeguards would affect the results you get with your prompt templates.
>
> Using our prompt templates, we did not observe any cases in which the safeguards were triggered, so we always got completion for the provided partial reference instance in the input prompt.
>
> > For unpaired instances, a random-length prefix is displayed in the prompt; how is this random length sampled? And what is its effect?
>
> The selection of both the instance itself and the length of the prefix used in the prompt was performed at random. As we also mentioned in our paper, for unpaired instances with multiple sentences, the prefixes are created by arbitrarily cutting the instances at the end of a complete sentence, whereas for instances containing a single (long) sentence, a random sentence fragment is removed.
> In terms of the effect of length, we would like to refer to the well-studied research by Carlini et al. (2023) where they found that memorization in LLMs significantly grows as we increase the number of tokens of context used to prompt the model.
>
> > Typo: "obtains" -> "obtained"
>
> Thank you! We fixed the typo in the current revised version of our paper.
>
> References: \
> [1] Quantifying Memorization Across Neural Language Models (Carlini et al., ICLR 2023)

---

### Official Review · Reviewer_KgsM · 2023-11-01

**Soundness:** 3 good
**Presentation:** 3 good
**Contribution:** 3 good
**Rating:** 8
**Confidence:** 3

**Summary:**

The paper offers a fresh perspective on assessing the capabilities of LLMs in terms of potential dataset contamination. The authors introduce two novel methodologies to measure these aspects. The first method uses BLEURT and ROUGE-L evaluation metrics, while the second leverages GPT-4's few-shot in-context learning prompt. A significant part of the evaluation revolves around identifying potential data contamination issues, and the results are compared against a baseline method, ChatGPT-Cheat. The findings underscore the nuances and intricacies in effectively evaluating LLMs and the paper serves as a guide to understand their limitations and potential.

**Strengths:**

Originality: The paper offers a fresh perspective on assessing the capabilities of LLMs in terms of potential dataset contamination. The methodologies introduced, especially the use of GPT-4's few-shot in-context learning, is innovative.
Quality: The research appears thorough with detailed evaluations using two different algorithms. The results are well-tabulated, and the comparison with ChatGPT-Cheat offers a clearer understanding of the proposed methods' effectiveness.
Clarity: The paper is structured coherently, and the methodologies, evaluations, and results are presented in a clear and organized manner, making it easier for the reader to follow.
Significance: Given the increasing utilization and reliance on LLMs in various applications, understanding their limitations and behaviors is crucial. This paper addresses this need, making it a significant contribution to the field.

**Weaknesses:**

Scope: The paper focuses primarily on GPT-3.5 and GPT-4. A broader range of LLMs could provide more generalizable insights.

**Questions:**

How do the proposed methods scale when evaluating even larger LLMs or when considering different architectures beyond GPT?
Could the authors provide insights into the trade-offs between the two algorithms, especially in terms of computational cost and reliability?

---

> ### Author Response · Authors · 2023-11-19
> **Official Response by Authors**
>
> Thank you for acknowledging the originality and significance of our study, and for your detailed review of our work.
>
> We address your comments and questions in the following in the same order they are listed.
>
> > The scope of LLMs.
>
> We appreciate your suggestion about including a broader range of LLMs. Our focus on GPT-3.5 and GPT-4 was informed by their prevalence in the field and the unique challenges of data contamination in closed-source LLMs, which are less transparent than open-source models both in terms of the model weights and pre-training data. While this focus allows for an in-depth exploration of these specific models, we acknowledge the potential benefits of expanding our study to include a more diverse set of LLMs for broader insights.
>
> > Detecting data contamination for larger LLMs using our method.
>
> For larger LLMs, based on the well-studied research by Carlini et al. (2023), as the memorization ability of LLMs significantly grows with their capacity, we believe for larger LLMs, it becomes even easier to replicate instances using our template prompts.
>
> > Detecting data contamination for architectures beyond GPT.
>
> Since our method hinges on replicating instances to validate their presence by emitting the inner learned knowledge, our method works best with decoder-only components (e.g., GPTs), although for other components like encoder-only (e.g., BERT), Magar & Schwartz (2022) investigated other methods.
>
> > Computational cost trade-offs between the two algorithms.
>
> In terms of the computational cost trade-offs between the two algorithms, Algorithm 1 with ROUGE-L is much friendlier than Algorithm 1 with BLEURT and Algorithm 2 (GPT-4 ICL), as these two need GPUs to run. On the other hand, the latter two methods use dense representations to evaluate the candidate text against the original instance, thereby performing better in terms of semantic match.
>
> > Reliability comparison between the two algorithms.
>
> In terms of reliability, since by using GPT-4 ICL, it is possible to provide examples of exact/near-exact matches from human evaluation for downstream judgments, it is much more aligned with human evaluations compared to Algorithm 1 with BLEURT and/or ROUGE-L.
>
> References: \
> [1] Quantifying Memorization Across Neural Language Models (Carlini et al., ICLR 2023) \
> [2] Data Contamination: From Memorization to Exploitation (Magar & Schwartz, ACL 2022)

---

### Official Review · Reviewer_6REU · 2023-11-01

**Soundness:** 3 good
**Presentation:** 3 good
**Contribution:** 3 good
**Rating:** 8
**Confidence:** 2

**Summary:**

The authors propose a method to detect dataset leakage/contamination in LLMs first at the instance level before they bootstrap to the partition (test, valid, train). At the instance level, they do so via a guided prompt that was crafted to bias the model towards outputting data in a format that is likely to overlap with the dataset example. Once a candidate dataset and partition is flagged, the authors mark it as leaked if either the overlap between reference instances is statistically significantly higher when using guided peompts compared to general prompts or is determined to be an exact or near match using a GPT-4 based classifier. The best classification models match with 92-100% accuracy the labels provided by human experts.

**Strengths:**

- Intuitive guided and general prompts to detect instance level contamination.
- Approximating human expert classification for exact and approximate match using GPT-4 as a classifier, i.e. approximating semantic match.
- Validation on a known contaminated LLM.

**Weaknesses:**

- The authors rely on the algorithm to begin with when deciding what partitions were not leaked and should be added during fine-tuning. This has a circular dependence/assumption. (This point was addressed during discussion with the authors as a writing/explanation issue rather than a true circular dependence).
- Different levels of data leakage is not considered. For example, would GPT-4 be detected as having seen paritions of datasets that follow well-known formats seen from other datasets if it sees only a metadata description of a new dataset? (This limitation is now acknowledged in a footnote with additional details on metrics as well as below in the discussion with the authors).

**Questions:**

My main question/concern is on the reliability of the instance level contamination detection. Specifically, if a dataset follows a well-known and observed in other dataset format, if a model such as GPT-4 sees the dataset description and meta-data, would it generate sufficiently many near matches to appear as contaminated with a new dataset, despite observing only metadata?
I understand that this work relaxes the prefix match from Sainz et al., but I wonder if this is likely to generate false signal in models that show an ability to generalise from few examplea and/or metadata.

---

> ### Author Response · Authors · 2023-11-19
> **Official Response by Authors**
>
> Thank you for recognizing the intuitive and innovative design of our approach, and for your valuable feedback.
>
> We address your comments in the following in the same order they are listed.
>
> > Circular dependence/assumption.
>
> To address the concern about circular dependence, we would like to mention that we were able to introduce contamination in our validation study (Section 3.3) using a set of synthetic instances from a fake dataset to break this circular dependency that seems to be present in the assumption. However, we found this a little bit confusing for the readers. Therefore, the main idea of contaminating using the partitions that were not leaked was only to maintain consistency throughout the paper for readers. Otherwise, there is no circular dependency.
>
> > Possible contamination through the metadata of another dataset.
>
> Regarding the possible contamination arising from metadata in another dataset—a point raised in both the Weaknesses and Questions sections—we would like to state that this is an interesting issue that we leave for future study. Our current work in this paper focuses on the complex issue of *detecting the presence* of contamination and not on *where* the contamination comes from. Nevertheless, we acknowledge the importance of identifying the sources of contamination.

---

> ### Comment · Reviewer_6REU · 2023-11-20
>
> > Circular dependence/assumption.
>
> It was indeed confusing between bootstrap and circular dependency, thank you for the clarification.
>
> > Metadata contamination
>
> I appreciate that this problem is beyond the scope of the paper, but still feel this should be mentioned as a potential limitation. I am still unconvinced if the paper can distinguish between contamination or just metadata contamination for a sufficiently accurate model. I do not feel this devalues the result, as for certain tasks, both can/should count as contamination regardless of the mechanism.

---

> > ### Author Response · Authors · 2023-11-22
> > **Official Response by Authors**
> >
> > > I appreciate that this problem is beyond the scope of the paper, but still feel this should be mentioned as a potential limitation. I am still unconvinced if the paper can distinguish between contamination or just metadata contamination for a sufficiently accurate model.
> >
> > Thank you for your reply. We appreciate your insightful suggestion regarding the inclusion of potential limitations. In response, we have added a new Limitations section to the paper, which is highlighted in blue on page 9. Due to space constraints, we added this section as a footnote. If extra space becomes available in the future, we will expand upon the Limitations section further. However, the current version addresses the key elements raised in your suggestion.
> >
> > Further, to thoroughly address your concern about distinguishing between contamination via the direct inclusion of dataset instances and metadata, we would like to refer to each of our proposed methods and the way in which each method detects contamination. In the following, we compare in detail the ability of each method to detect contamination via the direct inclusion of dataset instances or other means such as metadata.
> >
> > **ROUGE-L (Algorithm 1):** This metric measures only the lexical overlap similarities between generated completions and reference instances, without considering semantic relevance. Consequently, this method is incapable of detecting contamination other than verbatim contamination, which arises from the direct inclusion of dataset instances in the pretraining data of the LLMs. Therefore, this method cannot detect other forms of contamination, such as metadata contamination.
> >
> > **BLEURT (Algorithm 1):** Although this metric emphasizes semantic relevance and fluency when calculating overlap similarities between generated completions and reference instances, it still falls short in distinguishing near-exact and inexact matches.
> >
> > To illustrate, please consider the following example from the train split of the AG News dataset:
> >
> > **Instance:** Hunt for XP SP2 flaws seen in full swing Security experts said that while the new Service Pack 2 for Windows XP will bolster the operating system's security, hackers will still find a way to exploit any flaws. \
> > **Label:** 3 (Sci/Tech)
> >
> > When GPT-4 is prompted to replicate the second piece of this example given its random-length first piece, the generated completion is as follows:
> >
> > **First Piece:** Hunt for XP SP2 flaws seen in full swing Security experts said that while the new Service Pack 2 for \
> > **Second Piece:** ``Windows XP is a significant improvement, they expect hackers to be in full swing to find new vulnerabilities in it.`` (inexact match)
> >
> > As shown, while human judgment evaluates the generated completion as an **inexact match**, the BLEURT score reports a high overlap score of 0.78. As a result, while this method can capture some cases in which contamination occurs through means other than direct inclusion of dataset instances, it still fails to discern the correct sign of contamination in cases where exact replications are not possible due to factors such as metadata contamination or the probabilistic behavior of LLMs. To address this issue, we employ GPT-4 ICL.
> >
> > *Please note that all these completions and the overlap score can be replicated using the settings provided in the paper.*
> >
> > **GPT-4 ICL (Algorithm 2):** As shown and discussed above, even metrics such as BLEURT, which employ deep contextualized representations for computing overlap scores, struggle to recognize generated completions that human evaluation considers as near-exact or inexact matches with respect to the reference instances. GPT-4 ICL fills this gap by referencing a few representative examples from human judgments to determine the type of match with respect to the reference instance for downstream evaluation. Given the results reported in the paper, GPT-4 ICL outperforms the aforementioned methods using these representative examples by accounting for human evaluations, effectively handling near-exact and inexact cases. Therefore, GPT-4 ICL not only detects exact replicas stemming from the direct ingestion of dataset instances but also better aligns with human judgment in cases where exact replication is not possible due to factors such as metadata contamination or the probabilistic nature of autoregressive LLMs.
> >
> > In conclusion, considering GPT-4 ICL as our top-performing method, which others can employ for detecting data contamination based on this paper, our method does not distinguish between the various types of contamination, such as direct dataset instances inclusion and metadata contamination. In this regard, we encourage further research that not only detects contamination within LLMs but also determines its origins and the ways it manifests.
> >
> > Finally, we hope our response has effectively addressed your concern. We also appreciate the opportunity to discuss this topic in further detail in case you remain unconvinced by our response.

---

> > > ### Comment · Reviewer_6REU · 2023-11-23
> > >
> > > Thank you for the response and the addition of a limitations section (even as a footnote to point out and encourage further research). That was the main point I wanted to raise.
> > >
> > > Further, thank you for the insightful point-by-point discussion on each metric and how they (fail) to distinguish the source/type of contamination. As I agree this is beyond the scope of the paper, this explanation is beyond what I expected.
> > >
> > > I have adjusted my score to reflect my new confidence in the paper.

---

### Official Review · Reviewer_HqCE · 2023-11-02

**Soundness:** 2 fair
**Presentation:** 3 good
**Contribution:** 3 good
**Rating:** 6
**Confidence:** 4

**Summary:**

The paper investigates data contamination in Large Language Models (LLMs). To address this issue, the paper suggests a novel approach where a random-length initial segment of the data instance is used as a prompt, along with information about the dataset name and partition type. The paper then assesses data contamination based on the LLM's output. This evaluation can be done by measuring the surface-level overlap with the reference instance or by leveraging GPT-4's few-shot prediction capabilities.
Based on the results, the paper suggests that GPT-4 is contaminated with AG News, WNLI, and XSum datasets.

**Strengths:**

The proposed method is straightforward and adaptable to a wide range of datasets.

**Weaknesses:**

1. I have concerns regarding the soundness of the paper's evaluation methodology.
The proposed method hinges on the assumption that a data instance is contaminated in an LLM if the LLM can complete the instance based on its prefix. The paper's evaluation primarily revolves around how well the proposed methods are compared to human experts under this assumption However, these concerns raise doubts about whether the underlying assumption holds for several reasons.
(1) The inability of an LLM to complete a reference does not necessarily imply that the instance was not used during training. It could be attributed to model forgetting or the model's failure to memorize surface-level features while still having learned the semantic-level features of the data instance. This could lead to a high false negative rate in the evaluation.
(2) An LLM may have encountered the input of a data instance without having seen the actual data instance itself. For instance, in sentiment classification tasks, text can be included in the LLM's training set as long as its label is not provided alongside the text. The ability to complete the input text does not necessarily indicate data contamination in LLMs, potentially resulting in a high false positive rate.
(3) The unavailability of training data details for ChatGPT and GPT-4 due to their proprietary nature prevents a comprehensive evaluation of the proposed method. The current evaluation primarily focuses on how closely the proposed model aligns with human guess about data contamination, in cases where the actual training data is undisclosed. It seems essential to assess the method on models with publicly accessible training data, such as OpenLlama.
2. There's a potential fairness issue in using GPT-4 to evaluate the prediction from itself.

**Questions:**

See weakness

---

> ### Author Response · Authors · 2023-11-19
> **Official Response by Authors (1/2)**
>
> Thank you for your insightful comments on our paper, particularly regarding the soundness of our evaluation methodology. We appreciate the opportunity to address your concerns and provide further clarification on the underlying assumptions of our proposed method.
>
> Regarding your points 1.1 and 1.3, we would like to clarify that these concerns have already been considered and addressed in Section 3.3 of the paper. For point 1.2, we provide a detailed explanation below, complemented by an example for clarity. Moreover, to address your concern more thoroughly, we have included additional details in Appendix B in the updated version of our paper to show how infusing the labels in the input prompt accounts for false positives when generating downstream completions.
>
> We address your concerns in the following in the same order they are listed.
>
> > 1.1. The inability of an LLM to complete a reference instance.
>
> We understand that an LLM’s inability to complete a partial reference instance could be due to various factors. However, the memorization ability of LLMs is a well-studied topic. Carlini et al. (2023) suggest that memorization in LLMs significantly grows as we increase (1) the capacity of the model; (2) the number of tokens of context used to prompt the model; and (3) the number of times an example has been duplicated. The first two are the ones we **directly** make use of in our study. Therefore, consistent failure/inability to complete partial reference instances more plausibly indicates the lack of prior exposure to the data, rather than forgetting surface-level features or failure in memorization, especially in the context of LLMs such as GPT-4 and GPT-3.5.
>
> Further, in our study, we carefully examine **near-exact matches** to account for the exact scenarios you highlighted. To control the false negative rate in these cases, we empirically validated the number of near-exact matches that can signal partition-level contamination by performing controlled contamination on the base model of GPT-3.5 (Section 3.3). Then, based on these findings, we studied a set of numbers for both exact and near-exact matches as our **default setting** to generalize to the partition-level contamination.
>
> We would like to emphasize that the aforementioned numbers we studied as our default setting for determining partition-level contamination are **not fixed**. As mentioned in our paper, these criteria, i.e., the number of exact or near-exact matches, can be adjusted depending on a specific context and the perceived severity of contamination in a given setting. This flexibility in our approach allows for the effective addressing of potential false negative concerns. In other words, our replication-based strategy is completely separate from the way that a dataset partition is flagged as contaminated based on the number of exact or near-exact matches, adding the benefit of proactively handling potential false negative concerns.
>
> > 1.2. The importance of label inclusion in the input prompt.
>
> The pivotal idea behind incorporating the **exact label** in the input prompt is to exactly account for the false positive rates you are referring to. To address your concern, we intuitively show that when an LLM is instructed to complete a partial reference instance, the **exact label** corresponding to the reference instance is taken into account by the LLM for the downstream completion.
>
> To exemplify, in the WNLI dataset, sentence 1 entries are not unique, meaning that the same sentence 1 can be used with a different sentence 2, resulting in a different label. As a result, this is a suitable case to show how including a label can impact downstream completion. Please consider the following examples and completions:
>
> *Example 1 (from the **validation** split of the WNLI dataset, used in our paper in Figure 1):* \
> sentence 1: The dog chased the cat, which ran up a tree. It waited at the top. \
> label: 1 (entailment) \
> Completion for sentence 2 by GPT-4: `The *cat* waited at the top.` (exact match)
>
> *Example 2 (from the **train** split of the WNLI dataset, taken to support our claim here):* \
> sentence 1: The dog chased the cat, which ran up a tree. It waited at the top. \
> label: 0 (not entailment) \
> Completion for sentence 2 by GPT-4: `The *dog* waited at the top.` (this is an **exact match** with respect to the instance from the **train** split of the WNLI dataset, but an **inexact match** with respect to the instance from the **validation** split of the WNLI dataset (Example 1))
>
> *These completions can be replicated using the same settings provided in the paper.*
>
> As you can see, different labels lead to tailored completions, addressing your concern about false positives. We included additional details with more examples in Appendix B in the updated version of our paper to further emphasize the importance of label inclusion during the replication process.

---

> > ### Author Response · Authors · 2023-11-19
> > **Official Response by Authors (2/2)**
> >
> > > 1.3. The unavailability of training data for proprietary LLMs.
> >
> > We understand your concern about the unavailability of the pretraining data for proprietary LLMs. In this regard, one of the main purposes of our validation study (Section 3.3) is to address this concern by introducing contamination through **known data**  to the GPT-3.5 base model and replicating this known data using our approach to validate our strategy. We also experiment with the GSM8k dataset on GPT-4, as a known data included in the pretraining data of GPT-4 based on the technical report provided by OpenAI. These experiments showed that exact/near-exact matches, replicated through guided instruction, are indeed indicative of prior exposure to the data, thereby revealing data contamination. We have also provided several examples of exact/near-exact replicas from this experiment in Figure 2 and Table 11 in Appendix G in the updated version of our submission.
> >
> > In addition to the aforementioned validation study in our paper, we would like to refer to the study by Carlini et al. (2023) where they assessed the memorization ability of LLMs by replicating their training data. Inferring from their results and observations, replication of certain data confirms its presence in the training data of the LLMs. This implies that even **without direct access to the actual training data**, the presence of specific data can be inferred and confirmed through replication of this data by disclosing the LLM’s internalized information. From this perspective, as our method fundamentally detects contamination by confirming the presence of dataset instances through replication, experimenting with both closed-source and open-source LLMs would yield the same insights.
> >
> > > 2. Potential fairness issue in using GPT-4 to evaluate the prediction from itself.
> >
> > There seems to be some confusion about the potential bias in the GPT-4 ICL evaluation process. To clarify, we would like to emphasize that the replication and evaluation phases are carried out separately and independently. This means that during the evaluation process with GPT-4 ICL, there is absolutely no information about the source of the candidate text (generated by LLM) that is being assessed against the reference text. More importantly, all the evaluations from GPT-4 ICL are compared **against human evaluations** rather than being reported directly, mitigating potential fairness issues.
> >
> > Lastly, we appreciate the opportunity to discuss the potential of our work and are thankful for your valuable input. We hope our response has effectively addressed your concerns.
> >
> > References: \
> > [1] Quantifying Memorization Across Neural Language Models (Carlini et al., ICLR 2023)

---

> > > ### Comment · Reviewer_HqCE · 2023-11-20
> > >
> > > Thank you for the detailed response. I find that the table 6 and 11 along with the explanation in rebuttal addressed my concerns in general. I've adjusted the scores to reflect this.

---

### Author Response · Authors · 2023-11-19
**Official Message by Authors to All Reviewers**

We would like to extend our sincere appreciation to the reviewers for their dedication and thoroughness in evaluating our work. We greatly value the acknowledgment of the novelty and originality of our approach for detecting data contamination within LLMs and the significance of our contribution to the field given the potential major issues concerning data contamination. In the following, we present comprehensive responses to the comments from each reviewer. In response to the feedback, we have updated our submission, with a few modifications highlighted in blue, mainly to integrate the editorial suggestions made by the reviewers. Additionally, we have added a new appendix section to our paper—specifically, Appendix B—illustrating how incorporating labels in input prompts results in tailored completions and effectively addresses the concern raised regarding false positives.

During the rebuttal period, after our initial revision that entailed a few editorial adjustments and the addition of Appendix B, we received valuable input from one of the reviewers regarding adding a Limitations section. We recognized the value of this suggestion and proceeded to include this section alongside our previous adjustments. After trimming two lines containing a single word, consistent with our earlier modifications, these new modifications were added in blue within the paper on page 9.

---

### Meta-Review · Area_Chair_7qpm · 2023-12-13

**Metareview:**

This submission aims to study the data contamination problem in LLMs. The idea is simple:  to prompt an LLM with snippets from a dataset and analyzing the generated completions. Two strategies are used to evaluate the responses: 1) comparing the overlapping differences between prompts with/without the dataset specification and 2) using a GPT-4 based few-shot classifier.

One issue raised by the reviewer pertains to the evaluation of proprietary LLMs in the absence of access to their training data. However, this further underscores the significance of the studied problem --- examining the data contamination issue in (proprietary) LLMs.

All four reviewers recognize that the studied problem is important, the method is clear and simple, and the results are convincing. The consensus is reflected in the positive scores (8, 8, 6, 6) given to this submission, leading to a recommendation for clear acceptance.

**Justification For Why Not Higher Score:**

Debating whether to choose a spotlight or poster presentation. The issue studied is significant and the proposed approach is simple and easy to follow. A spotlight presentation could be valuable to emphasize the challenge of pre-training data in LLMs to a wider audience. However, a poster presentation would also be suitable.

**Justification For Why Not Lower Score:**

N/A

---

### Decision · Program_Chairs · 2024-01-16

Accept (spotlight)